

# Improving the efficiency of the Fukui trap as a capture tool for the invasive European green crab (*Carcinus maenas*) in Newfoundland, Canada

Jonathan A. Bergshoeff[1,2], Cynthia H. McKenzie[1,3] and Brett Favaro[1,2]

[1] Department of Ocean Sciences, Memorial University of Newfoundland, St. John's, Newfoundland and Labrador, Canada
[2] Centre for Sustainable Aquatic Resources, Fisheries and Marine Institute of Memorial University of Newfoundland, St. John's, Newfoundland and Labrador, Canada
[3] Northwest Atlantic Fisheries Centre, Fisheries and Oceans Canada, St. John's, Newfoundland and Labrador, Canada

## ABSTRACT

The European green crab (*Carcinus maenas*) is a crustacean species native to European and North African coastlines that has become one of the world's most successful marine invasive species. Targeted fishing programs aimed at removing green crabs from invaded ecosystems commonly use Fukui multi-species marine traps. Improving the efficiency of these traps would improve the ability to respond to green crab invasions. In this study, we developed four distinct trap modifications that were designed to facilitate the successful capture of green crabs, with the goal of improving the performance of the Fukui trap. We tested these modifications *in situ* during the summer of 2016 at two locations in Placentia Bay, Newfoundland. We discovered that three of our modified Fukui trap designs caught significantly more green crabs than the standard Fukui trap, increasing catch-per-unit-effort (CPUE) by as much as 81%. We conclude that our top-performing modifications have great potential for widespread use with existing Fukui traps that are being used for green crab removal efforts.

## INTRODUCTION

The European green crab, *Carcinus maenas* (Linnaeus, 1758) is a globally successful aquatic invader, now present on every continent with temperate shores (*Behrens Yamada, 2001*; *Roman, 2006*; *Darling et al., 2008*). In Newfoundland, the European green crab (hereafter green crab) was first detected in 2007, and it has since become established across the southern and western coasts of the island (*Best, McKenzie & Couturier, 2017*). These invasions threaten the native ecosystem through the destruction of sensitive eelgrass beds (*Malyshev & Quijón, 2011*; *Garbary et al., 2014*; *Matheson et al., 2016*), predation on native bivalves (*Ropes, 1968*; *Cohen, Carlton & Fountain, 1995*; *Matheson & McKenzie, 2014*; *Pickering et al., 2017*; *Poirier et al., 2017*), and competition with native species for

Corresponding author
Jonathan A. Bergshoeff,
jon.bergshoeff@gmail.com

food and habitat (*Cohen, Carlton & Fountain, 1995*; *Rossong et al., 2006*; *Rossong et al., 2012*; *Matheson & Gagnon, 2012*).

The complete eradication of an invasive species in a marine environment is virtually impossible once the organism has become established (*Bax et al., 2003*; *Lodge et al., 2006*), and may only be possible if the invasion is in a confined area and is addressed shortly after arrival (*Culver & Kuris, 2000*; *Simberloff, 2001*; *Bax et al., 2002*). In Newfoundland, the complete eradication of green crabs is no longer considered an option. Therefore, removal efforts have focused on trapping to supress invasive populations, and to slow further spread (*DFO, 2011a*). Focused trapping has become the predominant strategy for addressing green crab invasions on both the east and west coasts of Canada (*Duncombe & Therriault, 2017*; *Bergshoeff et al., 2018*).

These removal efforts usually use the Fukui multi-species marine trap (model FT-100, Fukui North America, Eganville, Ontario, Canada) to capture green crabs. These traps are favoured as they are light-weight, collapsible, durable, and can be deployed in large numbers from small boats or from shore. A standard Fukui trap consists of a rectangular, vinyl-coated high tensile steel frame ($60 \times 45 \times 20$ cm) covered with square, single-knotted, polyethylene mesh (12 mm bar length). There are two entrances at either end of the trap, where two netting panels form a horizontal "V" with a 45 cm expandable entry slit at the narrow end. To enter the trap through either of these entrances, green crabs must force themselves through the entrance which remains tightly compressed in its default position.

Our previous study was the first formal investigation of the interactions between green crabs and the standard Fukui trap (*Bergshoeff et al., 2018*). In that study, we mounted underwater video cameras to Fukui traps deployed *in situ* to assess the performance and efficiency of this gear, and to identify design features that were inhibiting green crab entry or facilitating exit prior to gear retrieval (*Bergshoeff et al., 2017*; *Bergshoeff et al., 2018*). Through these experiments, we discovered that only 16% of the green crabs that attempted to enter the Fukui trap were successfully captured. Our primary finding was that a combination of entanglement in the mesh and the restrictive trap entrance would often inhibit the successful entry of green crabs into the trap (*Bergshoeff et al., 2018*).

The main objective of this present study was to improve the performance of the Fukui trap as a capture tool for green crabs. Based on our video observations from the previous experiment, we developed four distinct modifications designed to facilitate the successful entry of green crabs into Fukui traps. We tested these modified Fukui traps *in situ* during the summer of 2016 and compared catch-per-unit-effort (CPUE) between each modified trap type and the standard Fukui trap. Our modifications were designed to be simple and practical, so that they could be easily applied to existing Fukui traps that are already in use for green crab removals. Our primary goal was to assess how these novel modifications perform, and to determine their potential for use in green crab removal programs that employ Fukui traps.
## METHODS

### Modifications

We developed four distinct trap modifications. For the first modification, we attached three 28.3 g (1-oz) lead bass casting sinkers with swivelling brass eyelets to the lower lip of each trap entry slit using 10.2 cm (4″) cable ties (Fig. 1A). The sinkers were evenly spaced along the entry slit, with one in the middle, and two attached 7 cm from the outer edge. The sinkers expanded the size of the trap entrance to approximately 2 cm at its widest point. For the second modification, we attached a 26 × 44 cm panel of black, fibreglass window screen (1 × 1 mm mesh size) to both the top and bottom of each trap entry tunnel (Fig. 1B). We used braided polyester string (0.825 mm diameter) to stitch these panels on top of the existing mesh. The panels were aligned so that a 4 cm wide strip of mesh extended through the trap entrance, which could be folded under the entry slit lip and stitched in place. For the third modification, we placed a thin strip of fibreglass window screen on the inside of the trap, adjacent to the entrance, which was designed to aid green crabs in pulling themselves through the restrictive opening (Fig. 1C). We cut a 4 × 54 cm strip of mesh and folded both ends to create a 42 cm long strip. The folded, reinforced ends were then attached to the inside walls of the trap using three 10.2 cm (4″) cable ties. Once attached, there was minimal slack left in the mesh strip, providing a flexible, yet stable surface for green crabs to grasp. The edge adjacent to the trap entrance was then loosely attached to the lower lip of the entry slit using three evenly spaced, partially-tightened cable ties. For the fourth modification, we used braided polyester string (0.825 mm diameter) to hold the trap entrance open (Fig. 1D). We used a 21 cm long piece of string to hold the upper and lower half of the entry slit open at the midpoint. The string was passed through either the top or bottom panel of the trap and tied to create a 7 cm loop. Once secured, the string created an oval-shaped opening, and increased the size of the trap entrance to approximately 6 cm at its widest point. For convenience, we named these trap modifications the *sinker*, *mesh*, *assist*, and *string* modifications, respectively, with an unmodified Fukui trap serving as our control.

### Fieldwork

We conducted our experiment in Fox Harbour, NL and North Harbour, NL during the summer of 2016 (Fig. 2). We ran the experiment for 26 days in total at Fox Harbour (June 15–18, June 21–25, June 28–July 5, July 12–16, August 4–12), and 9 days in total at North Harbour (July 18–27). We set traps in fleets of five, which consisted of the four different modified traps (i.e., *sinker*, *mesh*, *assist*, *string*) and an unmodified control trap. These fleets were deployed repeatedly in fixed locations across each study site, which we referred to as 'blocks'. The order of the five traps was randomized within each block. There were three blocks at Fox Harbour (FoxA, FoxB, and FoxC) and three in North Harbour (NorthA, NorthB, and NorthC) (Fig. 2). These blocks were spread out to provide sampling positions at multiple points within the two bays in which we conducted the experiment. The specific location of each block was selected based on accessibility by road and the presence of green crabs following a pilot study conducted in early June 2016 (North Harbour: June 2; Fox Harbour: June 8–9).

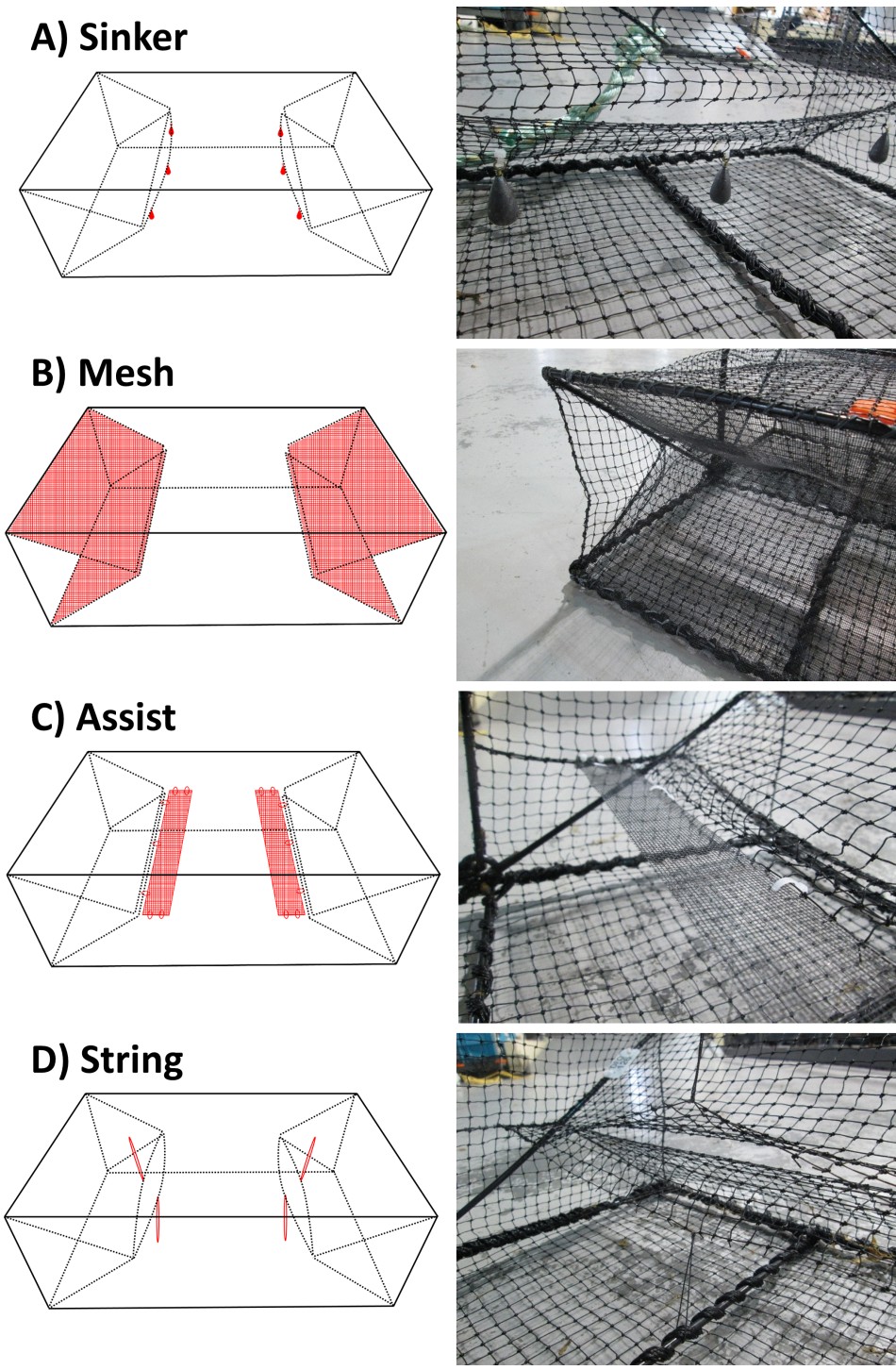

**Figure 1** **The four different trap modifications: sinker (A), mesh (B), assist (C), string (D).** The red coloured objects in each Fukui trap schematic indicate the modification features.

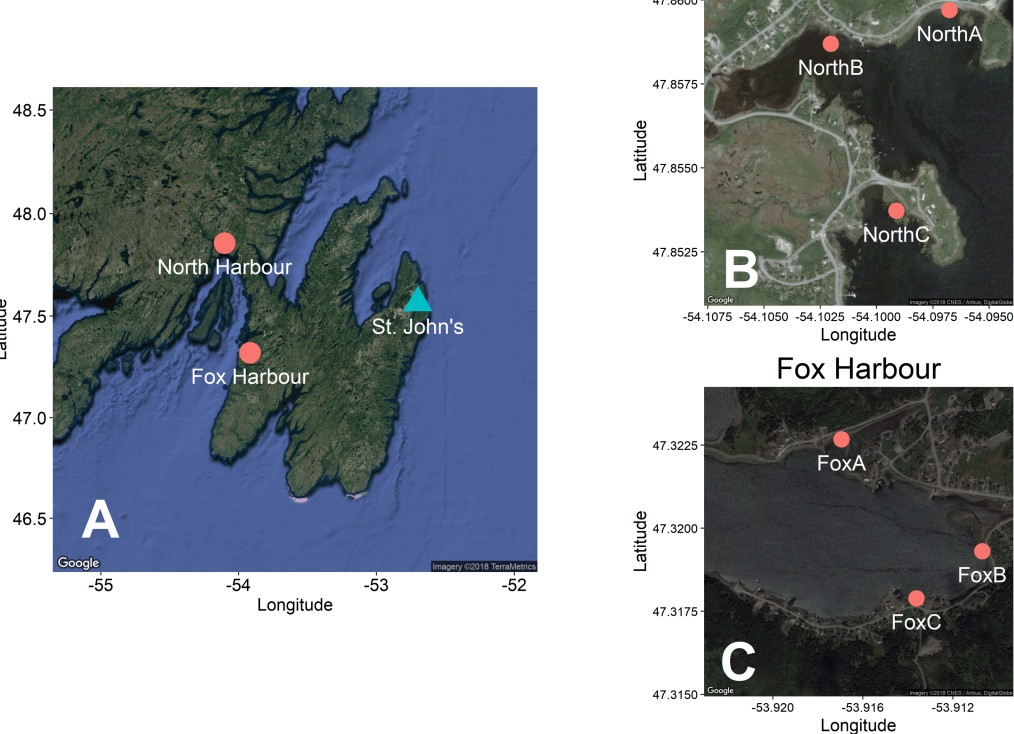

**Figure 2** **Maps showing the location of our study sites and experimental blocks.** (A) shows the location of North Harbour, NL and Fox Harbour, NL. The city of St. John's, NL is included for reference. (B–C) show the location of our experimental blocks at North Harbour and Fox Harbour, respectively. We produced these maps using the ggmap package (*Kahle & Wickham, 2013*) in R (*R Core Team, 2015*). Map A imagery ©2018 TerraMetrics. Map B and C imagery ©2018 CNES/Airbus, DigitalGlobe.

We baited each trap with approximately 200 g of Atlantic herring (*Clupea harengus*) that was thawed, cut into pieces, and placed in a perforated plastic bait container suspended inside the centre of the trap. We chose to use herring because it is the standard bait used by Fisheries and Oceans Canada (DFO) for green crab mitigation efforts (*Gillespie et al., 2007*; *DFO, 2011a*). Prior to each deployment, each trap was assigned a unique identification number and the order of traps within each fleet was pre-determined using a random sequence generator in R (*R Core Team, 2015*). We deployed the traps from shore during low tide so that they remained consistently submerged for the duration of the deployment. The traps within each fleet were placed approximately 10 m apart, matching the spacing used in other Fukui trap-based studies (*Gillespie et al., 2007*; *Gillespie et al., 2015*; *Behrens Yamada & Gillespie, 2008*; *Curtis et al., 2015*; *Duncombe & Therriault, 2017*; *Bergshoeff et al., 2018*). It is possible that the distance between adjacent traps could have had an influence on catch rates through bait attractant interference; therefore, we were careful to maintain consistent 10 m spacing between traps throughout the experiment. Our objective was to deploy each fleet for a 24-hour period, retrieving them at low tide the following day.

Upon retrieval of the traps, we placed all captured green crabs in large polyethylene bags (80 cm × 36 cm) along with a waterproof label indicating the unique identification number assigned to each trap. All bycatch species were visually identified to the lowest possible taxonomic level, recorded, and released as soon as possible. Once the catch was processed, the traps were baited with fresh herring, and re-deployed in a new random sequence. We repeated the entire process across all three blocks. All captured green crabs were euthanized by freezing, and prior to disposal they were counted, sexed, and measured. We measured the carapace width of each green crab using digital Vernier calipers between the fourth and fifth anterolateral carapace spines (i.e., notch to notch).

This project was approved as a 'Category A' study by the Institutional Animal Care Committee at Memorial University of Newfoundland as it involved only invertebrates (project # 15-02-BF), and our field experiment was conducted under experimental license NL-3271-16 issued by DFO.

## Statistical analysis
### Catch vs. trap type

We conducted all analyses and produced all figures using R Statistical Software (*R Core Team, 2015*). We employed a generalized linear mixed-effects model (GLMM) to test whether the number of green crabs captured per trap differed with trap type, and whether the deployment location (i.e., block) had an influence on catch. GLMMs are powerful statistical models that can be used to analyze non-normal data that involves random effects (*Bolker et al., 2009*). Our deployment durations were mostly consistent from one deployment to the next (mean = 23.9 h; SD = 2.4). Therefore, our CPUE was defined as the total number of green crabs captured per each individual trap deployment. To model catch as a function of the covariates, a negative binomial GLMM with a log link function was used (Eq. (1)). We had initially tested a Poisson GLMM, but found it to be overdispersed; therefore, we switched to a negative binomial distribution. The log link function ensured positive fitted values, and the negative binomial distribution was appropriate for our count data. We followed the equation nomenclature and style for presenting statistical models as outlined in (*Zuur & Ieno, 2016*).

The fixed covariates in our model are *trap type* (categorical with five levels: control, sinker, mesh, assist, string), and *block* (categorical with six levels: FoxA, FoxB, FoxC, NorthA, NorthB, NorthC). We had tested for an interaction between *trap type* and *location* (categorical with two levels: Fox Harbour, North Harbour), but found it to be non-significant; therefore, we removed it from our final model. In general, we found green crab distributions to be patchy, with a great deal of local-scale variation within each site. Therefore, *block* was included as a covariate to model the effect of spatial variation between the deployment locations. We deployed 15 traps per day and CPUE was not uniform across days; therefore, we included *study day* as a random intercept. This allowed us to incorporate the dependency structure among observations within the same study day, and to account for temporal variations in the environment (e.g., water temperature, weather).

To determine our final model we conducted stepwise backward model simplification, dropping non-significant terms (e.g., *duration*) one at a time until all terms in our model

were statistically significant (procedure outlined in *Crawley, 2012*). Our final model was specified as follows:

$$Catch_{ij} \sim NB(\mu_{ij}, k)$$
$$E(Catch_{ij}) = \mu_{ij}$$
$$var(Catch_{ij}) = \mu_{ij} + \mu_{ij}^2/k \tag{1}$$
$$\log(\mu_{ij}) = TrapType_{ij} + Block_{ij} + StudyDay_i$$
$$StudyDay_i \sim N(0, \sigma^2)$$

To fit the above model in Eq. (1), we used the lme4 package (*Bates et al., 2017*) in R (*R Core Team, 2015*). We verified the model assumptions by plotting residual versus fitted values, residuals versus covariates in the model, and residuals versus covariates excluded from the model.

### Carapace width vs. trap type

We constructed a linear mixed-effects model (LME) to test whether the mean carapace width of captured green crabs differed with trap type, and whether there was an interaction between carapace width for male and female green crabs (i.e., sex), and trap type. The carapace width measurements were normally distributed; therefore, we assumed a normal distribution in our model with an identity link (Eq. (2)).

The fixed covariates in our models were *trap type* (categorical with five levels: control, sinker, mesh, assist, string), *sex* (categorical with two levels: female, male), and *block* (categorical with six levels: FoxA, FoxB, FoxC, NorthA, NorthB, NorthC). *Block* was included in our model to account for any spatial variability between green crab populations in Fox Harbour and North Harbour. We tested for an interaction between *sex* and *trap type*. Finally, we included *study day* as a random intercept to incorporate the dependency among observations of the same study day, and to account for temporal variations. Our final model was specified as follows:

$$CarapaceWidth_{ij} \sim N(\mu_{ij}, \sigma^2)$$
$$E(CarapaceWidth_{ij}) = \mu_{ij}$$
$$var(CarapaceWidth_{ij}) = \sigma^2 \tag{2}$$
$$\sigma_{ij} = TrapType_{ij} + Sex_{ij} + Block_{ij} + TrapType_{ij} \times Sex_{ij} + StudyDay_i$$
$$StudyDay_i \sim N(0, \sigma^2)$$

To fit models in Eq. (2) we used the nlme package (*Pinhero et al., 2017*) in R (*R Core Team, 2015*). We verified the model assumptions by plotting residual versus fitted values. The residuals met the assumptions for homogeneity, normality, and independence.

## RESULTS

### Trapping effort and bycatch

We captured a total of 17,615 green crabs across 520 deployments (104 fleets) with an average catch of 34 green crabs per trap (SD = 25.4). We deployed 390 traps (78 fleets) in Fox Harbour, and 130 traps (26 fleets) in North Harbour (Table 1). We lost data from a single fleet (NorthA, $n = 5$ traps) in North Harbour on July 23 because the traps were

**Table 1** Summary of trap deployments at Fox Harbour, NL and North Harbour, NL.

|  | Traps deployed ($n$) | Replicates ($n$) | Deployment duration (h) (mean ± SD) | Catch per deployment (mean ± SD) | Min. catch | Max. catch | Total green crabs caught |
|---|---|---|---|---|---|---|---|
| Fox Harbour | 390 | 78 | 24.1 ± 2.2 | 35.5 ± 27.6 | 0 | 211 | 13,855 |
| North Harbour | 130 | 26 | 23.2 ± 2.8 | 28.9 ± 16.8 | 1 | 88 | 3,760 |
| Overall | 520 | 104 | 23.9 ± 2.4 | 33.9 ± 25.4 | 0 | 211 | 17,615 |

**Table 2** Summary of all bycatch species captured in each trap type. The number of green crabs caught in each trap type has also been included for comparison purposes.

|  | Control | Sinker | Mesh | Assist | String |
|---|---|---|---|---|---|
| Rock crab (*Cancer irroratus*) | 18 | 42 | 26 | 38 | 33 |
| Sculpin spp. (*Myoxocephalus spp.*) | 1 | 1 | 1 | 0 | 1 |
| Cunner (*Tautogolabrus adspersus*) | 0 | 1 | 1 | 1 | 1 |
| Rock gunnel (*Pholis gunnellus*) | 1 | 0 | 0 | 0 | 2 |
| Atlantic cod (*Gadus morhua*) | 1 | 0 | 1 | 1 | 0 |
| Winter flounder (*Pseudopleuronectes americanus*) | 0 | 0 | 0 | 0 | 2 |
| Sea trout (*Salmo trutta*) | 1 | 0 | 0 | 0 | 0 |
| Green crab (*Carcinus maenas*) | 2,738 | 4,023 | 3,529 | 4,778 | 2,547 |

washed ashore during a storm event. Deployment durations ranged from 13.8–27.1 h (mean = 23.9 h; SD = 2.4). Short deployment durations can be attributed to the logistical challenges of switching from trap deployments during evening low tide, to morning low tide.

Bycatch for both the standard and modified Fukui traps was minimal (Table 2). The most common occurrence of bycatch was rock crab (*Cancer irroratus*) in traps with the *sinker* and *assist* modifications. In total, we captured 157 rock crabs (mean = 0.3 rock crabs per trap; SD = 0.7) across all 520 deployments.

### Effect of trap type on CPUE

The actual number of green crabs caught in each trap type during our field experiment is summarized in Table 3. Our statistical model revealed that Fukui traps equipped with the *sinker* (GLMM: $\beta = 0.461$, S.E. $= 0.074$, $z = 6.220$, $p < 0.001$), *mesh* (GLMM: $\beta = 0.253$, S.E. $= 0.075$, $z = 3.370$, $p = 0.001$), and *assist* (GLMM: $\beta = 0.593$, S.E. $= 0.074$, $z = 8.030$, $p < 0.001$) modifications all caught significantly more green crabs than the unmodified control traps (Fig. 3). The catch rate was not significantly different between traps with the *string* (GLMM: $\beta = 0.029$, S.E. $= 0.075$, $z = 0.380$, $p = 0.705$) modification and the control traps. There was some variability in catch rates between the different experimental blocks, and the catch rate at FoxA was significantly higher than all other experimental blocks (Fig. S1). During the development of our model we found there was no significant interaction between *trap type* and *location*; therefore, despite spatial variations in catch between blocks, the CPUE of the modified traps was not influenced by the location.

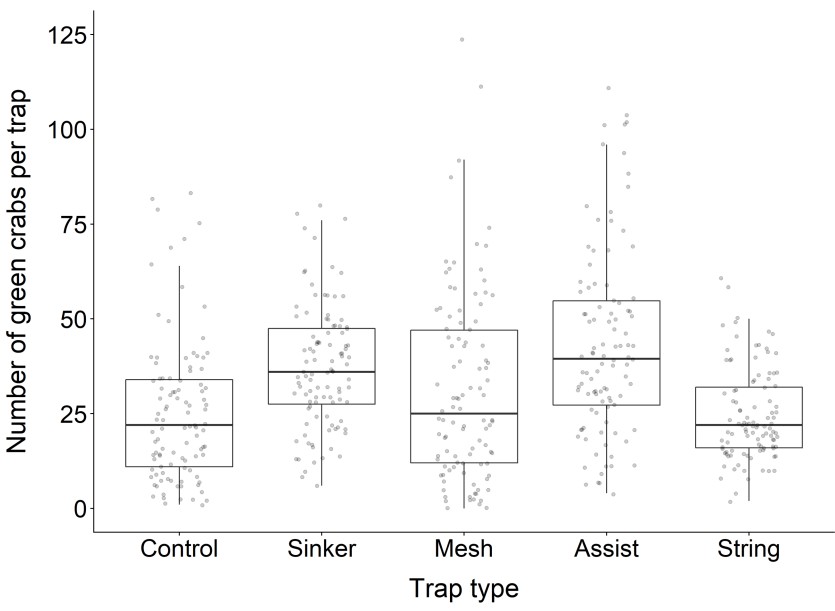

**Figure 3** **Boxplot illustrating the average number of green crabs captured in each trap type.** Each data point represents the number of green crabs captured for an individual trap deployment. The solid black line within each box depicts the median for that trap type. The lower and upper hinges of the box correspond to the first and third quartiles, respectively. The upper whisker extends to the largest value no further than 1.5 times the inter-quartile range (1.5*IQR), and the lower whisker extends to the smallest value no further than 1.5*IQR. Any data points beyond these whiskers are considered outliers. Outliers above 125 are not displayed ($n = 1, 1, 2, 2, 0$ respectively).

**Table 3** **Summary of green crab captured in each trap type.**

| Trap type | Traps deployed (n) | Catch per deployment (mean ± SD) | Minimum catch per deployment | Maximum catch per deployment | Total green crabs captured |
|---|---|---|---|---|---|
| Control | 104 | 26.3 ± 22.0 | 1 | 143 | 2,738 |
| Sinker | 104 | 38.7 ± 20.7 | 6 | 172 | 4,023 |
| Mesh | 104 | 33.9 ± 30.7 | 0 | 167 | 3,529 |
| Assist | 104 | 45.9 ± 30.7 | 4 | 211 | 4,778 |
| String | 104 | 24.5 ± 12.0 | 2 | 61 | 2,547 |

The output of our model is presented in Table 4, and the parameter estimates for each modified trap can be explained as follows: First, the *mesh* trap caught 1.29 (95% CI [1.11–1.49]) times as many green crabs as the control trap which translates to a 29% increase in catch relative to the standard Fukui trap. Second, the *sinker* trap caught 1.59 (95% CI [1.37–1.83]) times as many green crabs as the control trap which translates to a 59% increase in catch relative to the standard Fukui trap. Third, the *assist* trap caught 1.81 (95% CI [1.57–2.09]) times as many green crabs as the control trap which translates to an 81% increase in catch relative to the standard Fukui trap. Finally, the *string* trap did not show any statistically significant improvement over the control trap, catching 1.03 (95% CI

Table 4 **Estimated regression parameters, standard errors, z values, and P-values for the negative binomial generalized linear mixed-effects model (GLMM) presented in Eq. (1).** The estimated value of $\sigma_{\text{StudyDay}}$ is 0.368.

|  | Estimate | Std. error | z value | P-value |
|---|---|---|---|---|
| Intercept | 3.456 | 0.098 | 35.120 | <0.001 |
| Sinker | 0.461 | 0.074 | 6.220 | <0.001 |
| Mesh | 0.253 | 0.075 | 3.370 | 0.001 |
| Assist | 0.593 | 0.074 | 8.030 | <0.001 |
| String | 0.029 | 0.075 | 0.380 | 0.705 |
| FoxB | −0.323 | 0.066 | −4.920 | <0.001 |
| FoxC | −0.568 | 0.067 | −8.530 | <0.001 |
| NorthA | −0.527 | 0.173 | −3.050 | 0.002 |
| NorthB | −0.379 | 0.169 | −2.240 | 0.025 |
| NorthC | −0.333 | 0.169 | −1.970 | 0.048 |

[0.89–1.19]) times as many green crabs. This translates to a 3% increase in CPUE relative to the standard Fukui trap.

### Effect of trap type of crab size

We measured 17,598 green crabs in total (Fig. 4A). Across all trap types, carapace width for male green crabs ranged from 21.1–77.0 mm (mean = 52.4 mm; SD = 7.7). For female green crabs, carapace width ranged from 25.4–60.8 mm (mean = 41.8 mm; SD = 4.8). Male green crabs averaged significantly larger than female green crabs by 9.8 mm (LME: $\beta = 9.823$, S.E. = 0.369, $t = 26.636$, $p < 0.001$). For female green crabs, there was no significant difference in average carapace width between the modified traps and the unmodified control (Table 5). There was a significant interaction between *sex* and *trap type*. Both the *sinker* (LME: $\beta = 1.213$, S.E. = 0.486, $t = 2.494$, $p = 0.013$) and *string* (LME: $\beta = 1.914$, S.E. = 0.531, $t = 3.602$, $p < 0.001$) modifications caught male green crabs that were larger than the male crabs caught in the control traps. All parameter estimates, and the specific results of *block* can be found in Table 5. In general, green crabs caught in North Harbour were larger than green crabs caught in Fox Harbour, demonstrating the variability in green crab populations from one location to the next.

## DISCUSSION

### Modified trap performance

Modifications made to fishing gear designs can have a considerable impact on catch rates and catch composition. Modifications can be used to promote the switch to more sustainable fishing gears (*Ljungberg et al., 2016*; *Meintzer, Walsh & Favaro, 2018*), reduce the bycatch of non-target species (*Broadhurst, 2000*; *Furevik et al., 2008*; *Favaro, Duff & Côté, 2013*; *Serena, Grant & Williams, 2016*), improve selectivity for a target species (*Moran & Jenke, 1990*; *Boutson et al., 2009*; *Ovegård et al., 2011*; *Winger & Walsh, 2011*), and increase the overall catch rate of a fishing gear (*Sheaves, 1995*; *Nguyen et al., 2017*; *Meintzer, Walsh & Favaro, 2018*). In this study, we tested four different modifications designed to improve the efficiency of the Fukui trap as a tool for removing green crabs
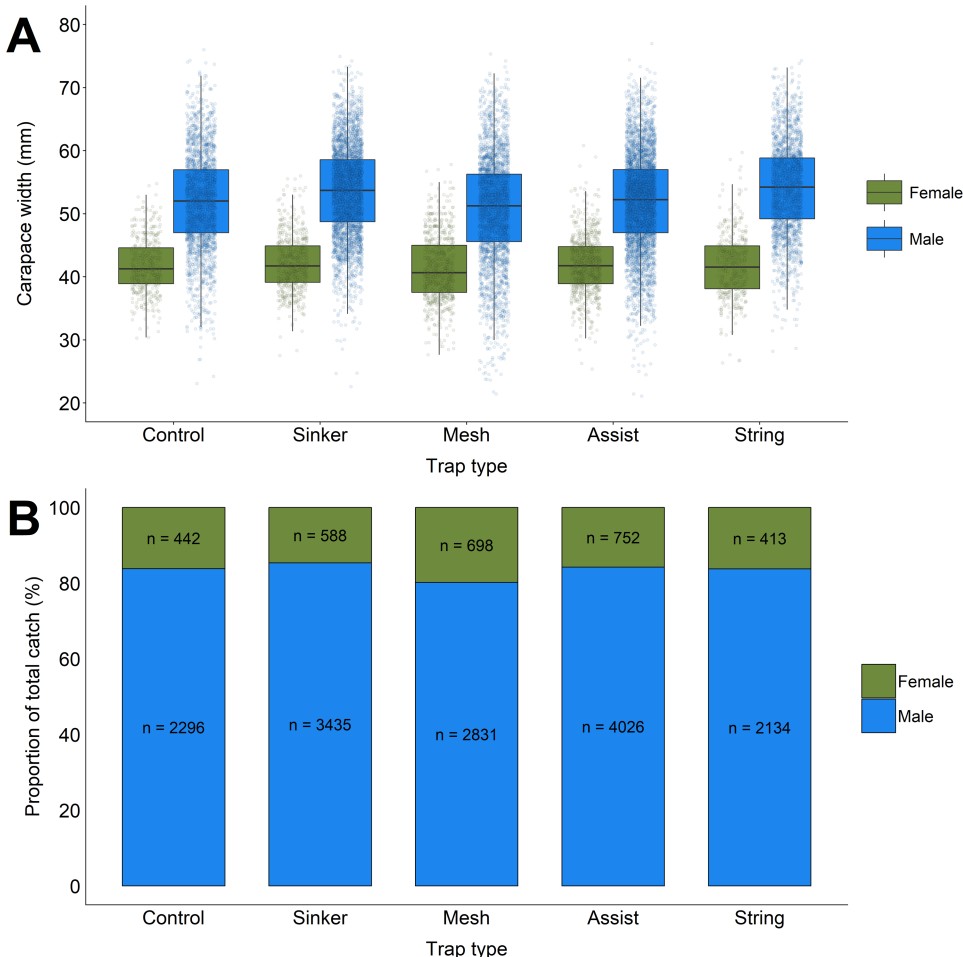

**Figure 4** **The mean carapace width (A) and total proportion (B) for male and female green crabs captured in each trap type.** Each data point in the boxplot (A) represents the carapace width of an individual green crab. The solid black line within each box depicts the median for that trap type. The lower and upper hinges of the box correspond to the first and third quartiles, respectively. The upper whisker extends to the largest value no further than 1.5 times the inter-quartile range (1.5*IQR), and the lower whisker extends to the smallest value no further than 1.5*IQR. Any data points beyond these whiskers are considered outliers. The bar chart (B) depicts the total proportion of female and male green crabs captured for each trap type. The numerical values represent the sample size for each proportion.

from invaded ecosystems. These modifications were specifically developed to address the most common inefficiencies in the design of the Fukui trap identified in (*Bergshoeff et al., 2018*)—primarily the entanglement of green crabs in the trap mesh, and the restrictive trap opening that inhibits the successful entry of green crabs into the trap.

Our least effective design was the *string* modification, which did not show any significant improvement in green crab CPUE when compared to the control. We suspect this low CPUE can be attributed to a high frequency of escape events. The *string* modification was designed to expand the trap entrance to approximately 6 cm at its widest point. By contrast, the control trap's entrance remains tightly closed in its default position. This

**Table 5 Estimated regression parameters, standard errors, *t* values, and *P*-values for the linear mixed-effects model (LME) presented in Eq. (2).** The estimated value of $\sigma_{StudyDay}$ is 1.223.

|  | Value | Std. error | *df* | *t* value | *P*-value |
|---|---|---|---|---|---|
| Intercept | 40.945 | 0.423 | 17,549 | 96.820 | <0.001 |
| Sinker | 0.464 | 0.447 | 17,549 | 1.039 | 0.299 |
| Mesh | −0.565 | 0.432 | 17,549 | −1.308 | 0.191 |
| Assist | 0.253 | 0.426 | 17,549 | 0.593 | 0.553 |
| String | −0.058 | 0.487 | 17,549 | −0.119 | 0.905 |
| Male | 9.823 | 0.369 | 17,549 | 26.636 | <0.001 |
| FoxB | 0.614 | 0.144 | 17,549 | 4.273 | <0.001 |
| FoxC | 1.695 | 0.154 | 17,549 | 11.004 | <0.001 |
| NorthA | 2.988 | 0.536 | 17,549 | 5.580 | <0.001 |
| NorthB | 2.190 | 0.521 | 17,549 | 4.207 | <0.001 |
| NorthC | 1.987 | 0.518 | 17,549 | 3.833 | <0.001 |
| Sinker:Male | 1.213 | 0.486 | 17,549 | 2.494 | 0.013 |
| Mesh:Male | −0.628 | 0.475 | 17,549 | −1.322 | 0.186 |
| Assist:Male | −0.300 | 0.464 | 17,549 | −0.646 | 0.518 |
| String:Male | 1.914 | 0.531 | 17,549 | 3.602 | <0.001 |

restrictive entrance does not allow green crabs to escape once captured; however; it also makes it harder for crabs to enter the traps (*Bergshoeff et al., 2018*). A large trap entrance facilitates entry, but can also increase the frequency of escapes, thereby reducing capture efficiency (*Archdale et al., 2007*). In our design, the strings caused both the upper and lower panels of the entry slit to curve outwards (Fig. 1D). We suspect this made it relatively easy for captured green crabs to climb out and escape, nullifying the benefits of a larger trap entrance. To mitigate this issue, our design could likely be improved by adjusting the length of the strings to create a smaller opening. However, despite the *string* modification being quick and easy to install, we found the overall design impractical, as the positioning of the strings made it difficult to clear captured green crabs from the trap once it was retrieved.

We designed the *mesh* modification to facilitate the entry of green crabs into the Fukui trap by preventing their pereopods and anterolateral carapace spines from becoming entangled in the trap mesh during entry attempts (*Bergshoeff et al., 2018*). We found the *mesh* modification was effective, capturing 29% more green crabs than the standard Fukui trap; however, we suspect that green crabs still encountered difficultly entering the trap. Although the *mesh* modification likely minimized entanglement, the fibreglass window screen panels overlaid on the existing trap mesh appeared to increase the tension of the entry slit. This likely made it more difficult for larger crabs to enter the trap (Fig. 4A). Furthermore, the slippery texture of the window screen may have made it difficult for crabs to gain enough traction to force themselves through the trap entrance. Our modification could likely be improved by replacing the window screen with a proprietary netting material designed for fishing gear. However, we found the process of stitching the mesh panels to the Fukui trap by hand was time-consuming, which could make adopting this modification impractical for large-scale green crab removal programs.

We designed the *sinker* modification to minimize the difficulty that green crabs experience when attempting to pass through the Fukui trap entrance (*Bergshoeff et al., 2018*). We found this modification to be simple and effective, producing catch rates that were 59% greater than the control. Unlike the *string* modification, which was also designed to increase the size of the trap opening, the *sinker* modification did not appear to facilitate frequent escape events. We suspect that the three 28.3 g (1-oz) sinkers expanded the lower half of the entry slit enough for green crabs to navigate through without much difficulty, but not enough that they were able to easily escape once captured, as inferred with the *string* modification. If we had chosen to use more weight, the trap entrances may have become too large, facilitating the escape of captured crabs. Although it is likely that some green crabs were able to successfully exit the trap, we suspect that most green crabs lacked the maneuverability to make their way back through once inside the trap. In the field, we found the *sinker* modification to be durable, withstanding repeated deployments for a cumulative total of 2,482.3 h without requiring any maintenance. Furthermore, unlike the *string* modification, traps equipped with the *sinker* modification were easily emptied upon retrieval. Finally, installing this modification took the least amount of time and effort, and could be used to quickly modify a large fleet of standard Fukui traps. These factors make the *sinker* modification a practical tool for capturing green crabs on a large-scale.

The *assist* modification was designed to aid green crabs in entering the Fukui trap, without increasing the size of the trap entrance. Our previous study revealed that even if a green crab was able to avoid entanglement and reach one of its pereopods or chelipeds through the entrance of a standard Fukui trap, there was nothing for it to grab hold of to pull itself through the opening. This lack of assistance would often result in green crabs failing to successfully enter the trap (*Bergshoeff et al., 2018*). Through the addition of the *assist* modification, we observed a dramatic 81% increase in the CPUE of the Fukui trap. However, despite the success of this modification in increasing CPUE, our current design would need improvements to make it practical for large-scale green crab removal efforts. Installation of the *assist* modification was tedious and time-consuming, and our prototype version lacked the durability required for repetitive usage and often had to be repaired over the course of our experiment. Like the *mesh* modification, the durability of the *assist* modification could likely be improved by replacing the window screen with a proprietary netting material designed for fishing gear. Alternatively, it is possible that some form of rigid plate could provide similar gains in CPUE. However, materials with a smooth surface should likely be avoided (e.g., a plastic plate), as these can be difficult for crustaceans to crawl over (*Miller, 1979*; *Favaro, Duff & Côté, 2013*). If the durability of the *assist* modification can be improved this design has the potential to greatly increase the efficiency of green crab removal efforts.

For female green crabs, there was no significant difference in carapace width between our modified traps and the standard Fukui trap. For male green crabs, traps equipped with the *sinker* and *string* modifications caught males that were larger than the unmodified control. These two modifications were the only designs where the size of the trap entrance was increased, which would explain their ability to capture larger green crabs. In general, female green crabs are smaller than male green crabs (*Best, McKenzie & Couturier, 2017*).

Therefore, the benefits of a larger entrance in both the *sinker* trap and the *string* trap are only realized by larger male green crabs, which likely experienced greater difficulty entering the *mesh*, *assist*, and control traps due to the restrictive trap entrance.

Removing large male green crabs from invaded ecosystems will likely have ecological benefits because reproductive success in males is directly related to size. During the mating season, male green crabs compete aggressively for access to receptive females for mating (*Berrill, 1982*). The most important factor that determines their success in these conflicts is their size (*Reid & Naylor, 1994*). These larger males have the reproductive advantage of bigger gonads and more spermatophores, which ultimately enables them to fertilize more eggs, or mate with a larger number of female green crabs (*Styrishave, Rewitz & Andersen, 2004*). Additionally, large crabs can forage to deeper depths in the sediment in search of food and can take larger prey (*Jensen & Jensen, 1985*; *Smith, 2004*). In general, these larger green crabs will have an advantage over native species when it comes to competition for food (*Cohen, Carlton & Fountain, 1995*). Therefore, there may be ecological benefits to removing the largest crabs from invaded areas, which is encouraged by our *sinker* and *string* modifications.

Furthermore, if future modifications can be refined to also increase the size of female crabs being caught this will have further ecological benefits, as larger females are capable of producing larger egg clutches and have higher reproductive success (*Audet, Miron & Moriyasu, 2008*; *Best, McKenzie & Couturier, 2017*). Under favourable conditions, a single adult female green crab can spawn up to 185,000 eggs per year (*Broekhuysen, 1936*; *Cohen, Carlton & Fountain, 1995*; *Grosholz & Ruiz, 2002*). Therefore, removing as many female green crabs as possible would have the largest per capita impact on population growth. In our experiment, the total catch composition for each trap type was mostly male, ranging from 80.2–85.4% (mean: 83.5%; SD = 1.94). We did not observe any meaningful variation in the total proportion of male and female green crabs caught in each trap type (Fig. 4B). The *mesh* modification caught a slightly greater proportion of female crabs compared with the other trap types; however, these minor selectivity gains are not enough to justify a shift in fishing strategy (e.g., using the *mesh* modification instead of the *sinker* or *assist* modification). Overall, we are trying to maximize the number of green crabs removed from invaded ecosystems. To accomplish this, we recommend using the *sinker* or *assist* modifications, which achieved the highest CPUE during our experiment and will likely have the greatest impact on the reproductive potential of invasive green crab populations.

## Bycatch

When compared with fishing gears such as bottom trawls and long lines, traps and pots are advantageous as they are often more selective for species and size, and they promote the live release of bycatch after the gear has been retrieved (*Suuronen et al., 2012*; *Poirier et al., 2018*). Our previous study in Newfoundland revealed that bycatch in the Fukui trap is minimal, and that most non-target species can be released unharmed upon retrieval of the trap (*Bergshoeff et al., 2018*). When targeting green crabs using the Fukui trap, native crustacean species are usually the most common form of bycatch (*Gillespie et al., 2007*; *Bergshoeff et al., 2018*). Modifications are often made to traps with the goal of reducing the

bycatch of non-target species, while maintaining the effectiveness of the gear at capturing target species (*Zhou & Shirley, 1997*; *Favaro, Duff & Côté, 2013*; *Serena, Grant & Williams, 2016*; *Poirier et al., 2018*).

The goal of our design modifications were to increase the overall capture efficiency of the Fukui trap; therefore, there was a risk that the modified traps would also increase the bycatch of non-target species. Fortunately, bycatch was minimal for all trap types, and mostly limited to native rock crabs (Table 2). Rock crabs were most commonly captured in traps equipped with the *sinker* and *assist* modifications. This is not surprising, as these traps were also the most effective at capturing green crabs. Our modifications were designed to facilitate the entry of green crabs into the Fukui trap; therefore, it makes sense that these modifications would also facilitate the entry of other crab species (i.e., rock crab). In both this study and our previous study, we did not observe any predation or mortality of rock crab bycatch (*Bergshoeff et al., 2018*). Overall, bycatch when using our modified Fukui traps in the coastal waters of Newfoundland appears to be of minimal concern. That said, the green crab is a worldwide invader. Therefore, prior to adopting these modifications in areas with a greater density of catchable bycatch species (e.g., along the west coast of Canada), they should initially be used with caution until the impact on native species is known.

## Practical application

We designed our trap modifications to be simple and practical, so that they could be applied to Fukui traps with minimal effort. Overall, based on durability and performance, we conclude that the *sinker* modification is an excellent option for improving the efficiency of the Fukui trap as a selective green crab capture tool. The *assist* modification demonstrated impressive performance; however, durability was an issue which currently limits its practicality for large-scale use. If these durability challenges are addressed, then this modification would be the most effective way to increase green crab CPUE within removal programs. However, the *sinker* modification has the added advantage of catching larger male green crabs. For both male and female green crabs, larger body size is associated with greater reproductive success and fecundity (*Kelley et al., 2015*). It has been shown that continuous trapping and the removal of larger crabs from an invaded area can cause a demographic shift towards a younger population, with reduced body mass and reproductive potential (*Duncombe & Therriault, 2017*). Furthermore, a reduction in the average carapace width of green crabs can cause a shift in their ecological role, from primary predators to potential prey for native crustaceans and shorebirds (*DFO, 2011b*). Therefore, even though the increase in average carapace width for the *sinker* modification is small, it could provide an advantage to green crab removal programs over time.

In this study, we chose to individually address the common inefficiencies of the Fukui trap; however, to further improve the trap's efficiency it may be possible to combine multiple design modifications into a single modified trap. For example, a Fukui trap equipped with the *sinker* and *mesh* modification would address both the restrictive trap entrance, and the issue of entanglement in the mesh. In addition to trap design modifications, future studies could investigate improved techniques for attracting green crabs to the trap itself.

The use of white and purple LED lights has been shown to significantly improve the catchability of snow crab (*Chionoecetes opilio*) traps (*Nguyen et al., 2017*). Similarly, it may be possible to increase the CPUE of Fukui traps using artificial lighting to attract green crabs. Furthermore, the type of bait used to attract a target species can also have an impact on CPUE (*Miller, 1990*; *Woll et al., 2001*; *Beecher & Romaire, 2010*; *Vazquez Archdale & Kawamura, 2011*). For green crab, it has been shown that Atlantic cod (*Gadus morhua*) and short-fin squid (*Illex illecebrosus*) can produce green crab catch rates that are statistically greater than Atlantic herring (*Butt, 2017*). When combined, these techniques could further increase the efficiency of the Fukui trap as a green crab capture tool.

Moving forward, it is essential that ongoing green crab removal programs continue to embed gear design studies within them. It would be a missed opportunity to incorporate the top performing modifications that we tested in this study into green crab removal efforts without further replicating the results against additional control traps. By replicating the results, it would be possible to quantify the benefits of making a switch to modified traps. When testing new modified Fukui traps, a portion of the traps that are being fished should always be devoted to controlled experiments to refine these modifications, until incremental innovation is no longer yielding benefits. For example, our *sinker* modification demonstrated a 59% increase in green crab CPUE; however, incremental innovations to this design (e.g., using both lighter and heavier sinkers) could pinpoint the optimal design that maximizes CPUE and green crab capture efficiency. Furthermore, by incorporating small-scale gear design studies within green crab management programs we can begin to build a culture around making invasive species fishing gear as efficient and effective as possible.

## CONCLUSION

In summary, this study demonstrates that dramatic improvements in the performance of the Fukui trap can be achieved through simple design modifications. Due to the widespread use and versatility of the Fukui trap as a green crab removal tool, it was important that these modifications were simple, durable, and effective. We conclude that the *sinker* modification meets all these requirements, and that existing Fukui traps can be easily retrofitted with this design for use on a large-scale. The *assist* modification is also an excellent choice for improving green crab CPUE; however, this design needs durability improvements before it is suitable for use in large-scale green crab removal programs. This emphasises the importance of embedding gear design experiments into ongoing green crab removal efforts. In doing so, incremental improvements can be made to determine the optimal gear design for improving green crab CPUE with Fukui traps. Intensive trapping using the standard Fukui trap has already proven to be an effective technique for reducing green crab populations (*Gillespie et al., 2007*; *DFO, 2011b*; *DFO, 2011a*). Therefore, we hope that our recommended design modifications will be adopted and refined by green crab removal programs as an efficient and selective tool to further reduce green crab populations in invaded ecosystems.

## ACKNOWLEDGEMENTS

We would like to thank all individuals who contributed to this project and are especially grateful to Mary Alliston Butt for her assistance with fieldwork. We also thank Maggie Folkins for her assistance in the field. We thank DFO for providing the Fukui traps that were used in this study. We thank Fox Harbour residents, Bob Buckmaster and Gerard O'Leary for their local knowledge and genuine interest in this research. Finally, we would like to respectfully acknowledge that we conducted this research on the unceded, unsurrendered ancestral lands of the Mi'kmaq and Beothuk, and the island of Newfoundland as the ancestral homelands of the Mi'kmaq and Beothuk.

### Funding

This project was funded by the Marine Environmental Observation Prediction and Response (MEOPAR) Early-Career Faculty Development Grant awarded to Brett Favaro et al. (EC1-BF-MUN). Jonathan A. Bergshoeff was supported by an Ocean Industry Student Research Award from the Research and Development Corporation of Newfoundland and Labrador (5404-1915-101). The funders had no role in study design, data collection and analysis, decision to publish, or preparation of the manuscript.

### Grant Disclosures

The following grant information was disclosed by the authors:
Marine Environmental Observation Prediction and Response (MEOPAR).
Early-Career Faculty Development Grant.
Ocean Industry Student Research Award from the Research and Development Corporation of Newfoundland and Labrador: 5404-1915-10.

### Competing Interests

The authors declare there are no competing interests.

### Author Contributions

- Jonathan A. Bergshoeff conceived and designed the experiments, performed the experiments, analyzed the data, prepared figures and/or tables, authored or reviewed drafts of the paper, approved the final draft.
- Cynthia H. McKenzie conceived and designed the experiments, contributed reagents/materials/analysis tools, authored or reviewed drafts of the paper, approved the final draft, provided supervision and advice throughout the study.
- Brett Favaro conceived and designed the experiments, analyzed the data, contributed reagents/materials/analysis tools, authored or reviewed drafts of the paper, approved the final draft, provided supervision and advice throughout the study.

### Field Study Permissions

The following information was supplied relating to field study approvals (i.e., approving body and any reference numbers):

This project was approved as a 'Category A' study by the Institutional Animal Care Committee at Memorial University of Newfoundland as it involved only invertebrates (project # 15-02-BF), and our field experiment was conducted under experimental license NL-3271-16 issued by Fisheries and Oceans Canada (DFO).

### Data Availability

Raw data are available in the Supplemental Materials.

### Supplemental Information

Supplemental information for this article can be found online at http://dx.doi.org/10.7717/peerj.6308#supplemental-information.

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
