# Peer review of "Improving the efficiency of the Fukui trap as a capture tool for the invasive European green crab (Carcinus maenas) in Newfoundland, Canada"

_PeerJ, doi:10.7717/peerj.6308_

## Round 0.1 · original submission · Minor Revisions

I apologize for the delay in making a decision on your manuscript. The reviews arrived along several revised manuscripts, creating a substantial backlog.

We ended up with three reviewers for your manuscript. All three concur that your is an interesting and useful study, well executed and clearly presented. I agree with this. It is one of the best-written manuscripts that I have encountered. They have provided a number of suggested minor revisions. I also noted a few minor issues when reading the manuscript.

Editor’s Comments:

L96 fourth (sp)
L172. I don’t think you defined the abbreviations in Eqn 1 and 2.
L202, 213. I find your Results sub-heads a bit awkward. They read more like a Methods section. Perhaps 3.1 could be called something like ‘Trapping effort and bycatch’, 3.2 could be eliminated, 3.2.1 could be changed to 3.2 and called something like ‘Effect of trap type on total catch’, and 3.2.2 could be changed to 3.3 and called something like ‘Effect of trap type on crab size’.
L216, 218, Tables 4, 5. I don’t think P can actually be zero, even if the program output gives this value. I think that p = 0.000 would be better changed to p < 0.001.
L225. Replace ‘is’ with ‘was’ for tense consistency.
L241. Replace ‘were’ with ‘averaged’ as the sentence is referring to the mean, not all individuals.
Fig. 2 caption, L3 ‘Maps B and C’.

Reviewer 1 ·

Basic reporting

I agree with the 4 points

Experimental design

I agree with the 4 jpoints

Validity of the findings

This detailed study aimed at improving the Fukui trap is very pertinent and timely and would be of great interest to the many researchers and managers around the world are using Fukui traps to monitor and manage green crab abundance in their invaded habitats.
The manuscript is well written and easy to follow. The experimental design, field execution and data analysis are well described. The reader gets a clear picture of which modification to the trap yielded the greatest improvement in catches. I recommend that this study be published.

Below are a few points the authors may want to consider.

• I was confused because the % increase in efficiency, as described in lines 226-235, does not always match a simple calculation of CPUE of modified traps to control trap as tabulated in Table 2. See table below. Am I missing something, or did the authors make a simple calculation error?

Trap Type Mean CPUE Table 2 -CPUE modification/control Increased Efficiency lines 226-235
Control 26.3
Sinker 38.7 38.7/26.3 1.47 59% - greater
Mesh 33.9 33.9/26.3 1.29 29% - matches
Assist 45.9 45.9/26.3 1.74 81% - greater
String 24.5 24.5/26.3 0.93 1.03 -- greater

• It might be helpful to the reader if the authors were to made some rough estimate of how much time it would cost to make the modifications. Sinker may be easy to attach. But they are not cheap and not that readily available in large quantities, while the mesh method, that requires sewing would be very time consuming.

• Starting with line 321 – continual trapping tends to preferentially remove large males from the system, as has been demonstrated by many researchers. But if it is done too well, it can have unintended consequences as Ted Grosholz found out in a closed system in Sea Drift Lagoon:
http://www.marinij.com/environment-and-nature/20170820/stinson-beach-green-crab-invaders-vex-scientists
In the absence of large males, young green crabs thrived, possibly indicating a lack of cannibalism and increased resources for the young crabs. ( – no need to include this in the manuscript, just something to keep in mind. )

Additional comments

This detailed study aimed at improving the Fukui trap is very pertinent and timely and would be of great interest to the many researchers and managers around the world are using Fukui traps to monitor and manage green crab abundance in their invaded habitats.
The manuscript is well written and easy to follow. The experimental design, field execution and data analysis are well described. The reader gets a clear picture of which modification to the trap yielded the greatest improvement in catches. I recommend that this study be published. Below are a few points the authors may want to consider.

• I was confused because the % increase in efficiency, as described in lines 226-235, does not always match a simple calculation of CPUE of modified traps to control trap as tabulated in Table 2. See table below. Am I missing something, or did the authors make a simple calculation error?

Trap Type Mean CPUE Table 2 -CPUE modification/control Increased Efficiency lines 226-235
Control 26.3
Sinker 38.7 38.7/26.3 1.47 59% - greater
Mesh 33.9 33.9/26.3 1.29 29% - matches
Assist 45.9 45.9/26.3 1.74 81% - greater
String 24.5 24.5/26.3 0.93 1.03 -- greater

• It might be helpful to the reader if the authors were to made some rough estimate of how much time it would cost to make the modifications. Sinker may be easy to attach. But they are not cheap and not that readily available in large quantities, while the mesh method, that requires sewing would be very time consuming.

• Starting with line 321 – continual trapping tends to preferentially remove large males from the system, as has been demonstrated by many researchers. But if it is done too well, it can have unintended consequences as Ted Grosholz found out in a closed system in Sea Drift Lagoon:
http://www.marinij.com/environment-and-nature/20170820/stinson-beach-green-crab-invaders-vex-scientists
In the absence of large males, young green crabs thrived, possibly indicating a lack of cannibalism and increased resources for the young crabs. ( – no need to include this in the manuscript, just something to keep in mind. )

Reviewer 2 ·

Basic reporting

No additional comments (see general comments).

Experimental design

No additional comments (see general comments).

Validity of the findings

No additional comments (see general comments).

Additional comments

PeerJ MS 29803

This is an interesting article addressing an important issue: the efficiency of traps used for green crab fishery/harvesting programs. Although the focus of the study is on a rather practical issue, it also offers a wealth of information for those interested on the ecology of these populations. The introduction and stated goals are very clear. The experimental design and the statistics used to assess the differences among traps are rigorous, and the results are clear and described to a good level of detail. Tables and Figures are informative and of good quality. Some of those results are frankly spectacular (80% increase of efficiency!), so the authors should be commended for conducting a well prepared study based on solid preliminary information. The Discussion highlights the main implications of the results and is not speculative. The practical implications of the results gathered by this study are obvious, and would be of great help for those working on the ecology of this invasive species or those trying to manage its invasions. I deemed the article a solid piece of research and I recommend acceptance with very minor changes (see below).

Specific comments

Lines 40-41: I suggest adding Pickering et al. 2017 (Mar Env Res) and Poirier et al. 2018 (Helgoland
Mar Res).
Line 43: I suggest adding Rossong et al. 2006 (JEMBE) and 2012 (Biol Inv).
Lines 93-96: If I understood the third modification correctly, please explain how tense that mesh was… in other words give us an idea about how “hard” or “stable” that floating “floor” was for the crabs crawling into the traps.
117-118: Briefly justify your choice of bait.
123-124: For the sake of the readers, briefly state whether you expect (or not) that spacing among traps to influence catch rates.
210-212: Provide a value (range?) of how many rock crabs were caught, to have a sense of how small the bycatch was.
289-291: It is not entirely clear what difference the sinker made (in comparison to the strings) to increase catch while reducing or maintaining escape rates low. Instead of altering escape rates, could the crabs use the sinkers somehow to help themselves to enter the traps and increase catch instead?
308-310: What about replacing that mesh with an acrylic or plastic plate or similar dimensions?
332-334: This statement is correct but is not justifiable given the results you report: you did not find larger females in the modified traps. I suggest you state “if future modifications also increase the size of females being caught… “
337-339 and 343-346: Although Poirier et al (2018; Fisheries Res) assessed a modification on a different gear (fyke net), I think is valuable to cite that work here as those results will make your results even more interesting (fukui bycatch is much lower).

·

Basic reporting

The purpose of the present paper is to evaluate modifications to the Fukui multi-species trap for removal of invasive European green crab in the waters of Newfoundland. The European green crab is a global invasive species and the Fukui trap is standard gear used in removal efforts in North America and elsewhere. These results will be useful to managers in many locales. At the same time, it is precisely because managers are likely to be interested in this work that I believe the authors should tread carefully in their recommendations (see below). The manuscript is well-written and clear and will make a valuable contribution.

The introduction & background provide adequate context but additional information (or at least citations) covering the invasion of green crab would be appropriate.

Experimental design

The research question is well defined, relevant & meaningful. I note the authors suggest their goal is to increase capture efficiency. However, capture efficiency is typically defined as the number of individuals caught as a proportion of the total number present. This is typically unknown. I would suggest the actual goal here is to increase catch per unit effort (CPUE).

The methods are described with sufficient detail & information to replicate.

Validity of the findings

Data are robust and statistical analysis is done appropriately. Conclusions are well stated (but see my thoughts below about contextualizing the results). It is important to note that these results pertain to the waters of Newfoundland. Managers/researchers using these modifications may see different results within their jurisdiction, particularly where bycatch is a concern.

Additional comments

Line 293-294: The authors suggest the sinker method is durable but this likely depends on the longevity of the traps. I know several researchers who have used the same Fukui traps for more than 15 years. The openings tend to sag over time and I’m afraid any extra weight may accelerate the process. I think any statements about durability should specify the number of deployment hours over which the modification was tested.

Line 336-356: The authors suggest the modifications do not increase bycatch. Again, I think it’s important to point out the context here given that green crab are a worldwide invader. While bycatch may be low in the waters of Newfoundland, conditions will be different elsewhere within the invasive range of green crab. In fact, on the US West Coast, researchers commonly restrict the size of the trap opening to reduce bycatch of demersal fish, sharks, and mammals.

Line 321-333 I agree that removing large adult green crab may be ecologically beneficial. However, female crab have a larger per capita impact on population growth (given the size of an individual brood, possible sperm storage, etc) and should be targeted for removal. I think it would be good to test for differences in catch of male and female crab across the four modified trap types. Moreover, I think figure 4 would be more useful if it explicitly considered total number of crab of each sex captured in each of the modified trap types.

---

## Round 0.2 · accepted · Accept

Thank you for the careful revision and rebuttal letter. I now consider the manuscript ready for publication.

#